# Bovine tuberculosis model validation against a field study of badger vaccination with selective culling

Graham C. Smith ®*, Richard Budgey ®

National Wildlife Management Centre, WOAH Collaborating Centre in Risk Analysis and Modelling, Animal and Plant Health Agency, York, United Kingdom

* graham.smith@apha.gov.uk

## Abstract

Bovine tuberculosis (bTB) is a costly disease in Britain and Ireland shared by cattle and badgers (*Meles meles*), and to reduce the infection in cattle to low levels some form of badger management is considered necessary. We compare the results of a badger field trial where test-positive badgers are culled, and test-negative badgers vaccinated (a TVR approach) with the results of the simulation model originally used to predict the effect of the trial in Northern Ireland. Initial model results depended strongly on whether social perturbation occurred in the badger population following culling, and the field study demonstrated no evidence for such behavior. Here we re-run the model with the initial conditions of the TVR study and with no social perturbation and predict a similar outcome in terms of number of badgers caught, number testing positive, and the substantial decline in prevalence. These results validate our model and demonstrate the utility of such predictive modelling for this disease system. This is particularly important as the UK government moves away from widespread badger culling in England toward more vaccination, as this combined approach of vaccination and selective culling based on test results may give a more robust method of disease management than just vaccination on its own.

## Introduction

In the British Isles, bovine tuberculosis (bTB: caused by *Mycobacterium bovis*) remains a costly disease shared by cattle and badgers (*Meles meles*), costing in excess of £150 million per year in England alone [1]. In the absence of management, it appears that both species could sustain bTB [2–4] in some locations although the frequency of spread between the two species is highly variable in different populations [5–8]. Thus some form of badger disease management would be required to reduce and retain bTB in cattle at very low levels. Various badger control strategies have been adopted: from reactive (reacting to detection in cattle herds) and localised

**Data availability statement:** All relevant data are within the manuscript and its Supporting Information files.

**Funding:** The author(s) received no specific funding for this work.

**Competing interests:** The authors have declared that no competing interests exist.

culling to large-scale culling in England with some vaccination, vaccination in Wales, and culling and vaccination in Ireland. The relative efficacy of these methods has been evaluated with simulation models [9–12], but culling approaches have generally been non-selective. Such culling risks behavioural perturbation of the badger population, which can induce increased ranging behaviour [13] and may increase disease prevalence in badgers and possibly in cattle [14,15].

Since 2010 the widely-used injectable vaccine, *Bacillus Calmette–Guérin* (BCG), became available for use in badgers and leads to a substantial reduction in disease in free-living badgers [16] and a degree of herd protection for cubs [17]. Field trials have confirmed that badger vaccination is not inferior to continuing culling [18]. Vaccination alone does not remove any (test positive) infected animals, so a combined policy may be more effective. By using trap-side tests to diagnose bTB (e.g., the dual path platform test: DPP), leads to the possibility of selective culling of test-positive animals and vaccination of test-negative animals. This is referred to as test and vaccinate or remove (TVR) approach. Selective culling may also be more publicly acceptable than widespread culling.

In Northern Ireland, where badger control had not previously been performed, an evaluation of this TVR approach was conducted. Initial modelling before the trial started suggested that the number of infected badgers remaining was very dependent on whether perturbation occurred: in the absence of perturbation a decline of about 70% in the number of infected badgers was seen, whereas with perturbation it was more modest [19]. Selective culling was also predicted to result in an 83% reduction in the number of animals culled [20]. With the completion of the subsequent five-year TVR study in Northern Ireland [21] we can re-examine these predicted effects on the badger population, and use the exact initial conditions in the field to validate the model output.

## Materials and methods

The TBi computer simulation model [12,20,22,23] was used to model the TVR study site in Northern Ireland. Input data included the initial population estimate, initial badger prevalence and the number of badgers captured each year [21]. Based on this, the model simulated the epidemiology, ecology and management of the badger population over the five-year course of the study to determine the population size and number infected. Estimates of annual disease prevalence during the trial, based on a Bayesian analysis combining multiple test methods [24], were used to validate the model output.

TBi is a stochastic, individual-based, spatially explicit model which simulates the life histories of a population of badgers at two-month timesteps. Life histories were generated using the probabilities of reproduction, mortality, dispersal, disease progression and disease transmission collected from the population at the Animal and Plant Health Agency's (APHA) Woodchester Park research station in Gloucestershire. Population density was taken from badger sett surveys conducted in County Down before the trial, and the demographic makeup of social groups was matched to the local population. The retention of some badger population parameter values from

the English model would have had minimal effect on the simulated output as the epidemiology is driven by badger density and disease prevalence, which were closely matched to the Northern Ireland study site. All model parameters and their source are given (S1 Table) and a full description of the model using the ODD protocol [25,26] (S2 File).

The model arena comprised of a 100 x100 grid cells, with each cell representing 200m x 200m; the total grid representing a 400 km² landscape area. In the central core of approximately 100 km² where badger management was undertaken, and the boundary was defined by the extent of participating farmland, there were 85 social groups and a population of about 550 badgers to match the field estimates [21]. Outside the core was a surrounding buffer two social groups wide where the possible influence of control could be observed and outside this any effect of culling was expected to be negligible. The grid was wrapped to form a torus to eliminate edge effects. Social groups were randomly distributed across the arena and all badgers were members of a group and occupied a territory which defined which social groups were neighbours.

## Characterisation of badgers

Individual badgers were characterized by the variables: social group, sex, age, and health-status. The age categories were cub, yearling (one-year old), and adult. The bTB-status categories were defined as: healthy, infected, single-site and multi-site excretor (*M. bovis* isolated from multiple body sites, e.g., sputum, faeces, urine, bite wounds) [27]. Probability of disease progression was based on field data from Woodchester Park [27]. Badger fecundity was density-dependent based on an upper limit of litters in each social group. Births were simulated at the start of the year, and litter size was modelled probabilistically from a distribution of known litter sizes [28], with a mean of 2.94 cubs per litter, and a sex ratio of 1:1. State-dependent mortality rates were based on field data from Woodchester Park [27]. Badgers up to two months of age (i.e., while still underground) had a higher mortality rate than older badgers [29]. Animals in the excretor disease classes also had higher mortality rates [27]. Badgers were allowed to disperse, usually to smaller social groups if available [30], based on sex-dependent probabilities (males more often than females) but independent of age and season. Badgers were also moved to neighbouring social groups in response to any demographic imbalance. The probability of bTB transmission between individual badgers was adjusted so the population disease prevalence matched the reported prevalence at the start of the study, estimated at 0.14 [24]. Disease transmission occurred between animals of the same and neighbouring social groups. As badgers interact more frequently with their own social group than with neighbouring groups, within-group transmission was given a greater probability (20-fold) than between animals in neighbouring groups [12]. Transmission probability increased as animals moved from excretor to super excretor class.

## Simulation of management operations

Prior to simulation of TVR management operations, the model was run for 100 years to allow the population and disease dynamics to stabilise after seeding. Management operations were simulated by allocating badgers a probability of capture based on the proportion of accessible land (0.94) supplied by DAERA (Menzies, F., pers. comm.) and trapping efficacy rates (0.54) [21]. Social groups were allocated to one of two trapping campaigns each year; territories not wholly within accessible land could still be subject to some level of control as badgers could be trapped away from the main sett. Badgers were individually marked during the study so recaptured animals were identifiable and this information was also available in the model.

In the field trial, animals were tested trap-side using the Dual-Path Platform VetTB test (DPP) on whole blood samples. In year one (2014), regardless of test result all badgers were vaccinated and released. In years two to five (2015–2018), test positive badgers were culled and test negative badgers vaccinated and released [21]. Field trial methods were replicated in the model with the simulation of one year of vaccination only, followed by four years of TVR. The model simulated DPP testing using a test sensitivity of 0.63 and specificity of 0.94 [24], based on the use of lines 1 or 2 in the DPP test under field conditions. In the field study, animals were vaccinated with BCG Danish in years 1–3 and BCG Sophia in years

4–5 due to supply issues. It was conservatively assumed in the model that both vaccines gave a 0.6 probability of providing full protection from infection for susceptible badgers [31]. Protection was for the lifetime of the badger, with further opportunity for full protection at subsequent capture for animals for which vaccination had previously been unsuccessful in providing protection. A simulation of the same population with no management was also undertaken to provide a baseline. A total of 100 simulations was run for each model scenario.

Field results suggest limited social perturbation resulted from badger removal operations [32,33]. Therefore, effects of perturbation were not simulated beyond the filling of demographic vacancies in neighbouring social groups described above. Although TVR does result in additional vacancies, there are many fewer than with non-selective culling and this demographic rebalancing contributes little additional transmission compared to the increased ranging behaviour seen in removal operations such as the Randomised Badger Culling Trial (RBCT) in England [34].

## Results

Model output is reported for the number of unique badger captures and the number of badgers testing positive using the simulated DPP test; these are compared to results from the study. Model output is also recorded for disease prevalence in the core, buffer, outer area of the arena and mean of the whole arena under TVR and no management; prevalence in the core is compared to the empirical estimate of prevalence from a Bayesian model combining three test methods used in the field trial: interferon gamma, culture and Dual-Path Platform [24].

The number of unique badger captures did not vary substantially over the course of the field study as relatively few were removed, and the population recovered before the following year. The model produced a similar result, although the number reported by the model was highest in the first year whereas in the field study the maximum occurred in the second year (Fig 1).

The simulated proportion of captured badgers that tested positive using the cage-side DPP test was in line with the general trend in population prevalence [21]. The number testing positive in the model was in reasonable agreement with the study in each year except 2016, where the field results were unusually low, but even in that year there was overlap

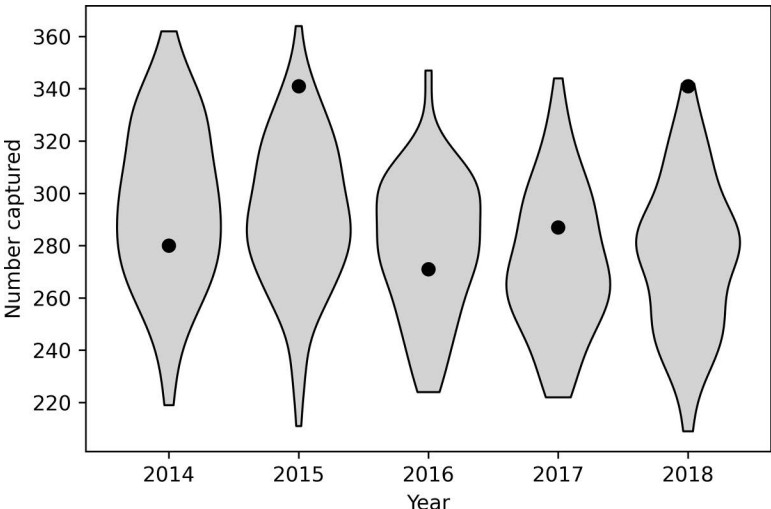

**Fig 1. Comparison between model results and data from the field trial for number of unique badger captures in each year of the trial.** Black dots indicate results from the TVR field trial and violin plots show the distribution of model predictions, with the width of each plot indicating the relative frequency of simulations with that output.

between model results and the 95% confidence limits of the empirical estimate (Fig 2). Since in year one (2014), all badgers were vaccinated and released regardless of test result, and in years 2–5 (2015–2018) test positive badgers were removed, for these later years the proportion testing positive is equal to the proportion of captured badgers removed.

A large benefit of TVR management was seen in the core where population level disease prevalence was predicted to reduce from the initial value of 0.14 to about 0.02, which closely matched the empirical estimate for change in annual prevalence [24] (Fig 3). The simulation was continued to year 2035 assuming no further management was applied (Fig 4). The model predicted prevalence in the core would slowly increase to about 0.05 some 15 years after management ended, although long-term model projections are always less reliable than short-term ones. The outcome when no management

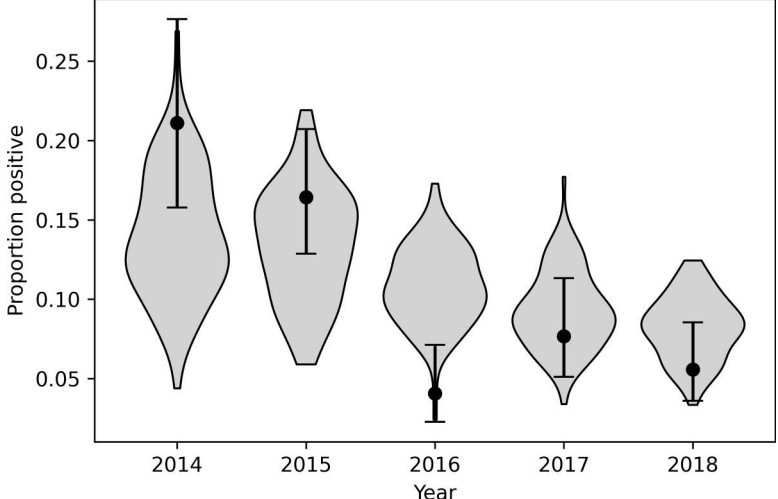

**Fig 2. Comparison between model results and data from the field trial for proportion of captures testing positive using the cage-side DPP test in each year of the trial.** Points indicate results from the TVR trial with error bars showing 95% binomial confidence interval, and violin plots show the distribution of model predictions.

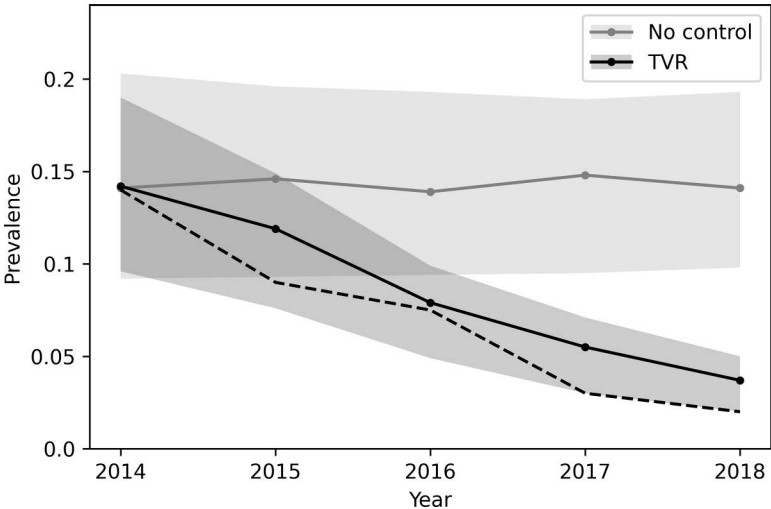

**Fig 3. Model results for median annual prevalence in the core area during the control period, with the no control scenario shown for comparison.** Shading indicates the inter-quartile range for the model predictions. The dashed line represents the estimated population-level prevalence from each year of the trial [24].

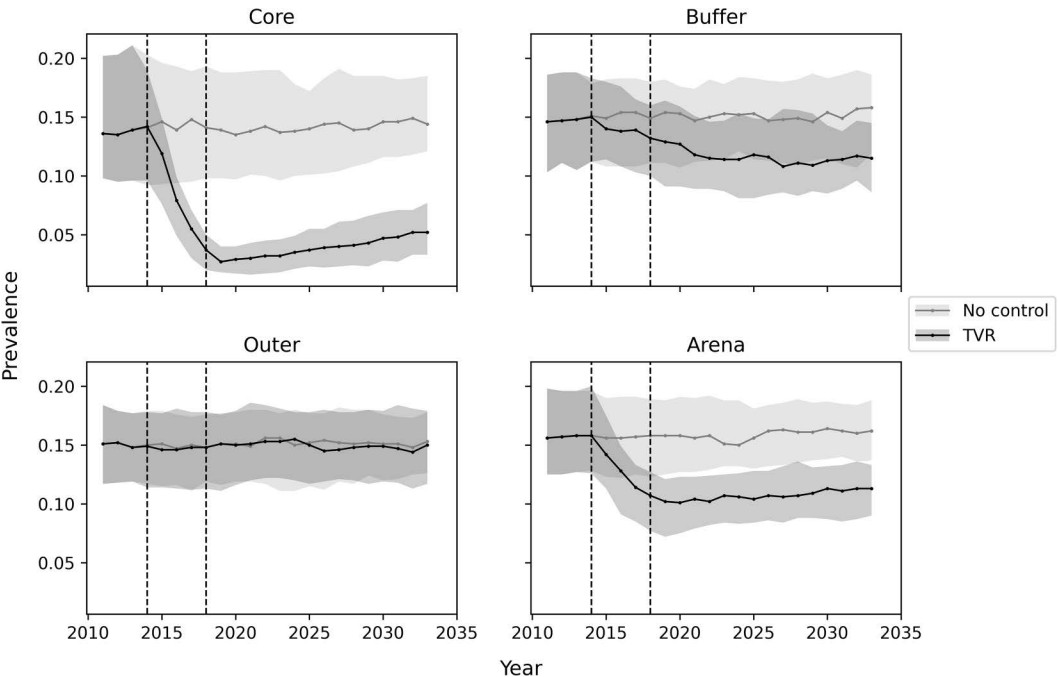

**Fig 4. Model results for median annual prevalence in each zone and overall simulated area (Arena), predicted to year 2035.** Vertical dotted lines indicate start and end of TVR trial. Shading indicates inter-quartile range for model predictions.

was applied was an unchanging population prevalence. A small benefit was seen in the buffer because some groups there may have partly overlapped with participating farms and therefore experienced limited removal/vaccination. There would also have been a small effect over the course of the study as some diseased animals in the higher prevalence buffer will have moved to the core, slightly reducing the benefit of control seen there and also some animals emigrated from the lower prevalence population in the core to the buffer, reducing the level of disease in the buffer.

## Discussion

In recent decades there has been an increasing reliance on using computer models to predict the consequences of disease outbreaks or disease control. Such models can rarely be validated prior to any management in the field. Here we take the original model used to evaluate a TVR badger management study for bTB in Northern Ireland and uniquely validate it against the data from the field trial. Such validation increases support for its use in other locations, or other scenarios.

During the five-year field trial a total of 824 badgers were caught, with between 271 and 341 unique captures each year [21]. This agrees well with the simulation, although the numbers caught in 2015 were higher than in 2014, whereas in the model the reverse was expected. Each year between 4% and 16% of badgers were removed: i.e., were DPP test positive [21]. This also agrees with our initial expectation of an 83% reduction in the number of badgers culled compared to a proactive cull: i.e., all trapped badgers would have been removed.

However, the most important prediction of the model was a substantial reduction in disease prevalence if social perturbation did not occur. During the trial a total of 105 individual badgers were followed using GPS collars, and there was no evidence of a change in home range size, neither annually, nor monthly, between the years of the study [32]. This strongly suggests that social perturbation did not occur in this population during the study. The field trial demonstrated a substantial

decline in prevalence during the trial [24], with the last years having a slightly lower prevalence than the simulated results, when we also assumed no social perturbation (Fig 3). Therefore, the model may be slightly conservative about the level of disease reduction. It is also worth noting that we predicted a slow recover of disease in the badger population, but it would require a repeated field study on this site to confirm or deny this longer-term prediction, and in general longer-term model predictions are less reliable as other factors may well occur in the interim.

Overall, this points to the success of the model in predicting the effects of the TVR approach in Northern Ireland. Since the simulated output depends most heavily on the badger social groups size and density, it therefore seems likely that the TVR approach would have similar outcomes across Ireland or other areas where badger dynamics are similar, but we cannot immediately extrapolate these results to England and Wales where social group size and density are both higher. However, in many areas in England badgers have been subjected to recent culling, and the density and social group size may now be more similar to the field study. We also did not account for any ongoing transmission from cattle to badgers throughout the trial and this would be expected to 'seed' the badger population with more *M. bovis*. Longer term model predictions are always less reliable than short term predictions, due to stochastic drift, changes in populations dynamics, habitat and farming, and the additional seeding that may occur from cattle would further erode the accuracy over time. Thus, we do not place any reliance on the longer-term dynamics of disease in the badger population at this stage. To gain more accurate longer-term dynamics would require linking this model to a cattle bTB model, and including cattle management.

## Conclusions

Models are often used to evaluate disease management scenarios in both animals and man. Such models are often fitted to field data to help parameterize them, but validation against unrelated field data is uncommon. We used an established simulation model of badgers and bovine tuberculosis and adjusted it to the local situation in Northern Ireland to predict the outcome of a proposed field study of a selective cull/vaccinate strategy. In this paper we report on the output of that model, after parameter changes to exactly fit the start of the field trial. The field study confirmed that no substantial social perturbation was apparent, and the revised model used to make retro-predictions that closely matched the real-world data. This is the first time that the simulation model was validated against badger vaccination and means that the use of the model for badger vaccination in other circumstances, such as in England where the focus is changing from culling to vaccination, should be reliable.

## Supporting information

**S1 Table. Input model parameters.**
(DOCX)

**S2 File. Model ODD protocol.**
(DOCX)

## Acknowledgments

The authors would like to acknowledge the help of Fraser Menzies for helpful discussion on both the original modelling for the TVR study, and the start conditions for this follow up analysis.

## Author contributions

**Conceptualization:** Graham C. Smith.

**Investigation:** Richard Budgey.

**Methodology:** Graham C. Smith.

**Project administration:** Graham C Smith.

**Software:** Richard Budgey.

**Supervision:** Graham C. Smith.

**Writing – original draft:** Graham C. Smith, Richard Budgey.

**Writing – review & editing:** Graham C. Smith, Richard Budgey.

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
