## [Decision Letter · Decision Letter 0]

Dear Dr. Smith,

Thank you for submitting your manuscript to PLOS ONE. After careful consideration, we feel that it has merit but does not fully meet PLOS ONE’s publication criteria as it currently stands. Therefore, we invite you to submit a revised version of the manuscript that addresses the points raised during the review process.

Please submit your revised manuscript by May 19 2025 11:59PM. If you will need significantly more time to complete your revisions, please reply to this message or contact the journal office at plosone@plos.org . A rebuttal letter that responds to each point raised by the academic editor and reviewer(s). You should upload this letter as a separate file labeled 'Response to Reviewers'.A marked-up copy of your manuscript that highlights changes made to the original version. You should upload this as a separate file labeled 'Revised Manuscript with Track Changes'.An unmarked version of your revised paper without tracked changes. You should upload this as a separate file labeled 'Manuscript'.

We look forward to receiving your revised manuscript.

Kind regards,

Frederick Quinn

Academic Editor

PLOS ONE

Reviewers' comments:

Reviewer's Responses to Questions

**Comments to the Author**

1. Is the manuscript technically sound, and do the data support the conclusions?

Reviewer #1: Yes

Reviewer #2: Yes

Reviewer #3: Yes

Reviewer #4: Yes

2. Has the statistical analysis been performed appropriately and rigorously?

Reviewer #1: N/A

Reviewer #2: Yes

Reviewer #3: Yes

Reviewer #4: Yes

3. Have the authors made all data underlying the findings in their manuscript fully available?

Reviewer #1: Yes

Reviewer #2: Yes

Reviewer #3: Yes

Reviewer #4: Yes

4. Is the manuscript presented in an intelligible fashion and written in standard English?

Reviewer #1: Yes

Reviewer #2: Yes

Reviewer #3: Yes

Reviewer #4: Yes

Reviewer #1: The paper is very well written and clearly shows that the model used in the paper is suitable for describing M. bovis prevalence in TB endemic population of Northern Ireland, and therefore could be applicable to England. Both the predicted data by the model and real data of the TVR in Northern Ireland show the benefit of vaccinating badgers and removing animals positive by DPP for decreasing TB prevalence in badgers.

Two minor points would help understand the paper better:

Line 129. Describe more precisely the parameters applied to the population without control.

Fig3: add the legend for the dotted line

In the discussion, could the decrease of prevalence observed in 2015 (before the removal of DPP positive badgers) be interpreted as the direct result of vaccinating the badgers? If so, wouldn’t it be interesting to include the modelling of the prevalence under a vaccination scheme only (without DPP removal), in parallel with TVR complete scheme, although this is not the main objective of the paper? Thank you.

Reviewer #2: This study presents a robust validation of a previously developed individual-based simulation model for bovine tuberculosis (TB) control in badgers, focusing on a real-world test and vaccinate or remove (TVR) field trial conducted in Northern Ireland. By tailoring the model to match the trial’s initial conditions and explicitly excluding social perturbation, the authors demonstrate a strong alignment between simulated and observed outcomes. The model accurately predicted trends in badger capture rates, test positivity, and a substantial decline in TB prevalence.

Questions and comments:

- Line 42. Could you highlight that BCG is 104 years old? I guess you meant to say that it became available to be used in badgers in 2010?

- Line 91. Could you define what do you mean by multi-site excretor, and how is that determined?

- The information provided in supporting information 2 document is great, I think some of it would be better suited to be added to the manuscript itself

Reviewer #3: Review of Bovine tuberculosis model validation against a field study of badger vaccination with selective culling

This manuscript describes the results of a model evaluating testing and selective culling of badgers in bTB affected zones with the actual field data. The model results were in close agreement with the field data, validating the model for use in these scenarios.

General comments:

The paper is well written and describes the use of the model and its’ predictions compared to field data. Generally, TB in cattle is referred to as bTB. Are the authors using only TB because of the inclusion of badgers? Since the bacteria is the same – Mycobacterium bovis – that bTB would be the appropriate acronym.

Specific comments:

Line 36 – what is meant by “reactive?”

Line 45 – suggest “Vaccination alone does not…”

Line 49 – suggest “Selective culling may also be more publicly acceptable…”

Line 79 – should this be 100kmX 100km grid?

Line 104 – I thought the Akhmetova paper didn’t find much badger to badger transmission and called into question the importance of them as a source for cattle?

Line 125 – suggest “In the field study,…”

Line 128 – what constitutes unsuccessful vaccination? The sentence sounds like these are badgers with multiple doses.

Line 144 – what three test methods were used?

Reviewer #4: Overall, this paper provides impressive evidence in support of the TVR activities on badger populations, the occurrence of perturbation, and M. bovis disease prevalence in an area of Northern Ireland, and possibly in other similar areas of the United Kingdom. One of the paper’s biggest strengths is its use of two previous, peer-reviewed studies, the initial modeling of M. bovis prevalence and the use of the results of a 5-year, TVR field trial in the area of interest in Northern Ireland for comparison to the revised model. I found this to strengthen the argument substantially. However, I also found this to be confusing in the manuscript, as multiple references are cited throughout when referencing the initial modeling or the field trial, and few details for the field trial and the initial modeling study are provided. Despite this, the article presents information that is useful in badger management methods and presents powerful modeling strategies for control of a zoonotic disease with important economical and public health considerations and I recommend its publication with the recommendations listed below. Great work!

Major Recommended Changes

In the Materials and Methods, a comparison of the field trial site (size characteristics, farms, etc.) and badger population details from the field trial to what was used in the model (arena area, 550 badgers and 85 social groups, etc.). (I did think the descriptions of the characterizations of the badger populations used in the model were very detailed and thorough.)

For the Results section, I would recommend an additional table for comparison of the results from the initial model, the field trial, and the new model in the Results Section as a Table 1, as none of these are provided outside the violin plots presented in the figures. A more detailed caption of the violin plots may also be helpful to describe the significance of the width of the plots. Labeling the data points within the figures may also be helpful for ease of reading the charts.

It would also be helpful to have a specific moniker for the initial model and field trial to indicate very obviously which is being referenced (vs “the study” (e.g. Line 142 and 146) which can be confusing and requires the reader to have to review the citations details). Including the entirety or relevant sections of those articles (+/- figures) in an appendix would also be helpful for referencing details that would be more helpful to the reader than citing those with the other references used in the article.

Minor Recommended Revisions

A figure of the model arena in the Materials and Methods showing the different designated core, buffer, and outer areas, with the farms included, would be helpful.

Line 26 – “… toward more vaccination.” recommend to edit to be more specific to vaccination and targeted culling based on testing results.

Line 32-34 – “In the absence of management (what kind of management?), it appears that both species could sustain TB (where? In the cattle population?) [2-4] although the frequency of spread between the two species is highly variable in different populations [5-8],” recommend revising to be more detailed, as is very early in the Introduction.

Line 71 – spell out APHA acronym for first use.

Line 73-76 – “The retention of some parameter values from the English model would have had minimal effect on the simulated output as the epidemiology is driven by badger density and disease prevalence, which were closely matched to the Northern Ireland study site.” Is this sentence needed if no parameter values were actually retained from the English model? (Is this the correct interpretation of this sentence?)

Line 81-82 – “The arena comprised a central core of approximately 100 km2 where badger management was undertaken…” recommend to edit to be more specific to TVR management done in the field trial.

Line 90-91 and – “ The TB-status categories were defined as: healthy, infected, single-site and multi-site excretor.” Recommend to add citation for these categories.

Line 101-102 – “The probability of transmission between individual badgers…” recommend to edit to TB transmission.

Line 107-108 – “Transmission probability increased as animals moved from excretor to super excretor class.” Recommend to add citation and definition for these categories (e.g. what is the difference between a super-excretor and a multi-site excretor (above)? Are they the same?)

Line 110 – “Prior to simulation of management operations…” recommend to edit to be more specific to TVR management.

Line 125-127 - It was assumed in the model that both vaccines gave a 0.6 probability of providing full protection from infection for susceptible badgers.” Recommend to add citation for vaccine efficacy.

Line 129, 143, 173, etc. – simultaneous use of “control” vs “management” to describe the specific TVR methods used the field trial and both models could be confusing to the reader in the context of experimental control and specific management technique.

Line 155-156 – “The simulated proportion of captured badgers that tested positive using the cage-side DPP test was in line with the general trend in population prevalence.” Recommend to add citation for this statement. Perhaps describe population prevalence in Materials and Methods section along with DPP Se/Sp and BCG vaccine efficacy?

**Do you want your identity to be public for this peer review?** For information about this choice, including consent withdrawal, please see our Privacy Policy

Reviewer #1: No

Reviewer #2: No

Reviewer #3: No

Reviewer #4: **Yes: ** Heather Martinez

---

## [Author Response · Author response to Decision Letter 1]

6 Jun 2025

All comments have been addressed in the attached document 'response to reviewers'.

---

## [Decision Letter · Decision Letter 1]

Bovine tuberculosis model validation against a field study of badger vaccination with selective culling

PONE-D-25-10268R1

Dear Dr. Smith,

We’re pleased to inform you that your manuscript has been judged scientifically suitable for publication and will be formally accepted for publication once it meets all outstanding technical requirements.

Kind regards,

Frederick Quinn

Academic Editor

PLOS ONE

Additional Editor Comments (optional):

Reviewers' comments:

Reviewer's Responses to Questions

**Comments to the Author**

Reviewer #3: All comments have been addressed

2. Is the manuscript technically sound, and do the data support the conclusions?

Reviewer #3: Yes

3. Has the statistical analysis been performed appropriately and rigorously?

Reviewer #3: Yes

4. Have the authors made all data underlying the findings in their manuscript fully available?

Reviewer #3: Yes

5. Is the manuscript presented in an intelligible fashion and written in standard English?

Reviewer #3: Yes

Reviewer #3: All previous comments have been addressed. Why is there a minimum character count in this section???

**Do you want your identity to be public for this peer review?** For information about this choice, including consent withdrawal, please see our Privacy Policy

Reviewer #3: No

---

## [Editor Report · Acceptance letter]

PONE-D-25-10268R1

PLOS ONE

Dear Dr. Smith,

I'm pleased to inform you that your manuscript has been deemed suitable for publication in PLOS ONE. Congratulations! Your manuscript is now being handed over to our production team.

Kind regards,

on behalf of

Dr. Frederick Quinn

Academic Editor

PLOS ONE